# Revolutionizing Neuroimmunology: Unraveling Immune Dynamics and Therapeutic Innovations in CNS Disorders

**DOI:** 10.3390/ijms252413614

**Published:** 2024-12-19

**Authors:** Corneliu Toader, Calin Petru Tataru, Octavian Munteanu, Razvan-Adrian Covache-Busuioc, Matei Serban, Alexandru Vlad Ciurea, Mihaly Enyedi

**Affiliations:** 1Department of Neurosurgery, “Carol Davila” University of Medicine and Pharmacy, 020021 Bucharest, Romania; corneliu.toader@umfcd.ro (C.T.); razvan-adrian.covache-busuioc0720@stud.umfcd.ro (R.-A.C.-B.); matei.serban2021@stud.umfcd.ro (M.S.); prof.avciurea@gmail.com (A.V.C.); 2Department of Vascular Neurosurgery, National Institute of Neurology and Neurovascular Diseases, 077160 Bucharest, Romania; 3Department of Opthamology, “Carol Davila” University of Medicine and Pharmacy, 020021 Bucharest, Romania; 4Central Military Emergency Hospital “Dr. Carol Davila”, 010825 Bucharest, Romania; 5Department of Anatomy, “Carol Davila” University of Medicine and Pharmacy, 050474 Bucharest, Romania; mihaly.enyedi@umfcd.ro; 6Neurosurgery Department, Sanador Clinical Hospital, 010991 Bucharest, Romania; 7Medical Section, Romanian Academy, 010071 Bucharest, Romania

**Keywords:** neuroimmunology, neuroinflammation, central nervous system, blood-brain barrier, immune modulation, microglia, astrocytes, CNS tumors, precision medicine, immunotherapy

## Abstract

Neuroimmunology is reshaping the understanding of the central nervous system (CNS), revealing it as an active immune organ rather than an isolated structure. This review delves into the unprecedented discoveries transforming the field, including the emerging roles of microglia, astrocytes, and the blood–brain barrier (BBB) in orchestrating neuroimmune dynamics. Highlighting their dual roles in both repair and disease progression, we uncover how these elements contribute to the intricate pathophysiology of neurodegenerative diseases, cerebrovascular conditions, and CNS tumors. Novel insights into microglial priming, astrocytic cytokine networks, and meningeal lymphatics challenge the conventional paradigms of immune privilege, offering fresh perspectives on disease mechanisms. This work introduces groundbreaking therapeutic innovations, from precision immunotherapies to the controlled modulation of the BBB using nanotechnology and focused ultrasound. Moreover, we explore the fusion of immune modulation with neuromodulatory technologies, underscoring new frontiers for personalized medicine in previously intractable diseases. By synthesizing these advancements, we propose a transformative framework that integrates cutting-edge research with clinical translation, charting a bold path toward redefining CNS disease management in the era of precision neuroimmunology.

## 1. Introduction

### 1.1. The Intersection of Immunology and Neuroscience

Neuroimmunology, positioned at the vibrant crossroads of neuroscience and immunology, has transformed over recent decades, redefining our understanding of the central nervous system (CNS) as a domain actively engaged with the immune system. Once considered as “immune-privileged”, the CNS was long thought to operate in isolation from peripheral immune activity, shielded by the blood–brain barrier (BBB) [1]. However, a wealth of groundbreaking discoveries has illuminated a profound, dynamic partnership between neural tissues and immune components. No longer seen as merely defensive, the immune system within the CNS now appears to be integral to the stability, adaptability, and resilience of neural networks [2].

Central to this revised understanding are microglia—the CNS’s innate immune cells—which have been recognized as key regulators of brain health. Microglia operate as vigilant sentinels, continually scanning the neural landscape and responding not only to injury or pathogens, but also to everyday maintenance demands [3]. These cells play a critical role in clearing cellular debris, remodeling synapses, and even guiding neurogenesis. During early brain development, microglia engage in synaptic pruning, selectively eliminating unnecessary connections to refine neural circuits. Intriguingly, emerging evidence suggests that microglia respond to neuronal signals, adjusting their activity based on neural cues and subtly shaping brain function in real time [4].

Astrocytes, once considered to be passive support cells, are now recognized as essential partners in CNS immunity. Working in tandem with microglia, astrocytes maintain BBB integrity, modulate the brain’s biochemical environment, and release signaling molecules that influence both local and systemic immune responses [5]. They contribute to a dynamic surveillance system within the CNS, adapting to changes and actively protecting the neural landscape [6]. Recent studies show that astrocytes can produce cytokines and chemokines, adding to their roles in neuroprotection and bridging communication with peripheral immune cells. These insights position astrocytes as pivotal coordinators, balancing neural activity and immune response within the CNS [7].

Perhaps one of the most surprising findings in recent neuroimmunology is the discovery of lymphatic vessels within the brain’s meninges. This revelation overturned the longstanding belief that the brain was immune-isolated, demonstrating a direct pathway for immune cell migration and fluid exchange between the brain and the peripheral immune system [8,9]. Meningeal lymphatics may play a crucial role in clearing metabolic waste and amyloid-β, a process with profound implications for neurodegenerative diseases like Alzheimer’s. The discovery of these lymphatic pathways has spurred a new wave of research into how immune cells enter and exit the CNS and how immune surveillance contributes to CNS health [10].

Together, these findings reveal the CNS as a highly immune-active organ, where the partnership between neurons, glial cells, and immune components is essential to maintaining stability and health. The interplay between immune cells and the neural environment now forms a core aspect of CNS function, shedding light on mechanisms that can be harnessed for therapeutic benefit in an array of neurological diseases.

### 1.2. Scope and Objectives of the Review

This paper delves deeply into the most recent developments in neuroimmunology, exploring how immune mechanisms protect, adapt, and occasionally disrupt CNS function. Through a close examination of cutting-edge research, we aim to unravel the diverse ways that immune processes shape CNS resilience and response. Key areas of focus include the cellular architecture underlying immune surveillance within the brain, the intricate signaling pathways that modulate inflammatory responses, and the ways that immune activity influences conditions such as cancer, vascular disease, and neurodegeneration.

To begin, we look at the cellular and molecular foundations of CNS immune surveillance, detailing how the microglia, astrocytes, and meningeal lymphatic pathways work together to maintain neural homeostasis. By understanding the coordination between these components and their interactions with peripheral immune cells, we gain a nuanced view of CNS immunity as a continuously adaptive system. Special attention is given to the impacts of age and genetic predispositions, exploring how these factors can disrupt immune balance and predispose the CNS to disease.

The discussion then turns to how immune mechanisms contribute to the development and progression of specific CNS disorders, with a focus on cancer, neurovascular disease, and neurodegenerative conditions. The phenomenon of immune escape in brain tumors, for example, reveals how certain tumors evade immune detection within the CNS, allowing unchecked growth. This understanding opens up new possibilities for immunotherapies targeting gliomas and other brain cancers. Additionally, we examine the role of chronic inflammation in neurodegenerative diseases like Alzheimer’s and Parkinson’s, where prolonged immune activation is being increasingly recognized as a driver of neurodegeneration. Insights into these processes are paving the way for therapies that aim to moderate harmful inflammation, slow disease progression, and even prevent neuronal loss.

Emerging therapeutic strategies are a focal point of this exploration, particularly immune-modulatory drugs, advanced pharmacological agents designed to cross the BBB, and neuromodulation techniques that target neuroimmune pathways. Breakthroughs in BBB-penetrating technologies, including nanotechnology and biologically engineered carriers, have created opportunities to deliver drugs directly to CNS targets, expanding the treatment possibilities for neurodegeneration, brain cancer, and injury. Techniques like transcranial magnetic stimulation (TMS) and focused ultrasound (FUS) are also gaining attention for their ability to influence the immune activity in localized brain areas, offering new approaches to manage neuroinflammation.

Our objectives in this analysis are threefold. First, we aim to illuminate how immune mechanisms underpin CNS health and resilience, examining the dual role of immune cells in maintaining brain function and responding to injury or disease. Second, we investigate the evolving landscape of therapeutics that target neuroimmune pathways, including advances in immune-modulating agents, BBB-penetrating drugs, and non-invasive neuromodulation techniques. Finally, we explore future directions for research, highlighting areas of great potential that may redefine CNS disease management. Through a synthesis of recent findings, this discussion provides a unique perspective on the regulation of neuroimmune interactions and their promising implications for treating neurological disorders.

This work is particularly focused on bridging foundational concepts with the latest therapeutic innovations and potential future breakthroughs. By connecting cellular and molecular insights with practical applications, our discussion aims to present a thorough and forward-thinking view of neuroimmunology. This exploration of CNS immunity redefines the traditional boundaries within neuroscience and immunology, highlighting a rapidly evolving field that promises transformative approaches to treating brain disorders. With an eye on precision medicine, the work emphasizes neuroimmune pathways as critical targets, envisioning an era of tailored, patient-specific therapies that address the unique immune and neural profiles of individuals affected by neurological diseases.

## 2. Immunology and the Nervous System

### 2.1. Overview of the Immune System in a Neurological Context

Within the CNS, immune functions are uniquely tailored to meet the demands of a highly specialized and sensitive environment. Unlike peripheral immune systems, which are designed to respond vigorously to external threats, the CNS’s immune activity operates with precision, balancing defense mechanisms with the need to protect delicate neural structures [11]. This tailored approach relies heavily on innate immunity, which provides fast, non-specific responses customized to the CNS’s requirements. A cornerstone of this system is the role of microglia, the resident immune cells of the CNS. Microglia are dynamic and versatile, maintaining constant surveillance of neural tissues, prepared to respond to both endogenous changes and exogenous threats. These cells are not only defenders, but also crucial contributors to CNS stability, as they clear cellular debris, regulate synaptic connections, and foster neurogenesis when required [12].

Microglial adaptability in the CNS extends beyond conventional immunity. Equipped with receptors that sense molecular patterns associated with pathogens and cellular stress, microglia respond to a broad spectrum of signals, initiating precise actions suited to CNS needs. During early brain development and synaptic remodeling phases in adulthood, microglia exhibit activity that resembles fine-tuning, selectively “pruning” superfluous synapses to refine neural networks [13]. This synaptic sculpting is now understood as a critical component of healthy brain function, influenced by microglial interactions with neurons and astrocytes, which facilitate communication between these cells [14]. These roles position microglia as integral regulators, far beyond their traditional designation as immune responders, highlighting their essential contribution to CNS homeostasis [15].

Astrocytes, another major cell type in the CNS, are being increasingly recognized as key players in its immune landscape. Beyond their supportive roles in neurotransmission and BBB maintenance, astrocytes actively participate in immune signaling, forming a responsive communication network with microglia and, when necessary, with peripheral immune cells [16]. Astrocytes produce a variety of cytokines and chemokines, which are critical for controlling inflammation within the CNS and serve as mediators that can either amplify or dampen immune responses. This functional flexibility allows astrocytes to act as dynamic regulators, adjusting local immune environments in response to shifts in neural activity or metabolic demand [17]. Their influence extends to maintaining BBB integrity, with astrocytic end-feet enveloping blood vessels and modulating barrier permeability based on neural and immune cues, an essential function for CNS protection [18].

While adaptive immune responses remain limited under normal conditions to protect sensitive CNS structures, adaptive cells like T cells and B cells can enter the CNS under specific circumstances, such as in response to infections or injury [19]. Their entry is tightly regulated and transient, enabled only when acute threats necessitate their involvement. This restriction reflects the CNS’s need to prevent immune overactivity that could compromise neuronal function. Interestingly, recent studies show that, when adaptive immune cells do enter the CNS, they engage in highly specialized interactions with resident cells, often with guidance from chemokine signals produced by astrocytes and endothelial cells. This tightly controlled system of innate and occasional adaptive responses ensures that CNS immunity remains robust yet balanced, tailored to protect without excessive collateral damage [20].

### 2.2. Blood–Brain Barrier and Immune Cell Trafficking

The BBB serves as the CNS’s primary line of defense, a selectively permeable boundary composed of endothelial cells, pericytes, and astrocytic processes that tightly regulate the entry of immune cells and other molecules. This structural barrier is essential for maintaining the CNS’s immune privilege, preventing unnecessary immune activation within the sensitive neural environment [21]. Under normal conditions, the BBB restricts immune cell access, admitting only select molecules and, in rare cases, specific immune cells under highly regulated processes. However, this barrier is not static; it dynamically adjusts its permeability in response to stress, inflammation, or injury, a feature that underscores its adaptive complexity [22].

When systemic infections or trauma occur, the BBB responds by upregulating specific adhesion molecules, such as ICAM-1 and VCAM-1, on its endothelial surface. These molecules enable immune cells to dock onto the BBB, allowing controlled entry into the CNS, where they can address localized threats [23]. Chemokine gradients further guide these immune cells, directing their movement once they cross the BBB to precisely target affected areas. For example, chemokines like CXCL10 and CCL2, produced by astrocytes and endothelial cells, create a directional signal that ensures that immune cells concentrate in regions requiring intervention while leaving healthy neural regions undisturbed [24,25]. This selective immune trafficking demonstrates the sophistication of BBB regulation, where only the necessary immune responses are permitted to cross into CNS territories.

However, in pathological conditions, the BBB’s regulatory capacity can be compromised. In multiple sclerosis (MS), for instance, the barrier’s integrity becomes weakened, allowing for the excessive infiltration of immune cells that perpetuate chronic inflammation [26]. This breakdown initiates a self-sustaining inflammatory loop in which infiltrating immune cells release cytokines that further degrade BBB tight junctions, creating a continuous cycle of immune cell entry and inflammation. Emerging research is now focused on identifying molecular targets within the BBB to prevent these breaches without suppressing immune function entirely [27]. For example, targeting specific pathways that regulate BBB permeability could allow therapies to block harmful immune cell entry in conditions like MS while preserving the barrier’s protective properties [28]. Additionally, recent advances in pharmacology are exploring methods to selectively modulate the BBB, temporarily increasing its permeability to facilitate drug delivery directly to CNS targets, a promising approach for treating CNS diseases that traditionally resist pharmacological intervention [29].

### 2.3. Neuroinflammation: Mechanisms and Implications

Neuroinflammation represents a dual-edged response within the CNS, offering essential protective actions in acute scenarios while posing risks when it becomes chronic. In the context of acute injuries or infections, neuroinflammation mobilizes resident immune cells like microglia and astrocytes, activating rapid responses to contain pathogens, clear cellular debris, and initiate tissue repair [30]. This immediate response involves the release of cytokines such as IL-1β, TNF-α, and IL-6, which coordinate immune actions to limit damage and support healing processes. A crucial aspect of this response is the resolution phase, wherein anti-inflammatory mediators, including IL-10 and TGF-β, signal immune cells to stand down once threats are neutralized [31]. This resolution process is essential for returning the CNS to a balanced state, emphasizing neuroinflammation’s inherent self-regulatory nature [32].

However, neuroinflammation’s role shifts dramatically when it becomes chronic, a state that often emerges in neurodegenerative diseases like Alzheimer’s and Parkinson’s. In these conditions, the ongoing activation of microglia and astrocytes leads to prolonged pro-inflammatory signaling, creating a toxic environment that endangers neuronal health [33]. Chronic neuroinflammation is characterized by sustained cytokine production, reactive oxygen species (ROS) generation, and the release of other neurotoxic factors, which collectively contribute to neuronal dysfunction [34]. Microglial “priming”—a state in which microglia become sensitized and over-responsive to minor stimuli—is a significant contributor to this chronic state. Primed microglia are particularly reactive to neurodegenerative triggers, such as amyloid-β (Aβ) or α-synuclein, which amplify inflammatory responses and drive neurodegeneration [35].

Recent research has shed light on the mechanisms underpinning microglial priming, with epigenetic modifications emerging as potential culprits. Studies indicate that prior exposure to inflammatory events can alter the epigenetic landscape of microglia, enabling them to “remember” these events and respond with a heightened sensitivity to subsequent stimuli [36]. This immune memory is beneficial in peripheral immunity, where quick responses to familiar threats are advantageous. However, in the CNS, where immune responses must remain tightly controlled, primed microglia contribute to pathological inflammation. For example, in Alzheimer’s disease, primed microglia produce elevated levels of IL-1β and TNF-α upon encountering amyloid plaques, intensifying neuroinflammation and accelerating neuronal damage [37]. This phenomenon underscores the need for therapies that specifically target primed microglia, offering a path to alleviate chronic inflammation without suppressing the beneficial, acute responses required for CNS repair [38].

A critical focus in neuroimmunology is the development of therapies that can break this cycle of chronic neuroinflammation without compromising protective immune functions. Selective inhibitors that target key pro-inflammatory cytokines or pathways in primed microglia are one avenue of exploration, as they could reduce harmful inflammation while preserving the CNS’s innate healing capabilities [39]. Additionally, compounds that promote the CNS’s resolution phase—by enhancing anti-inflammatory mediators such as IL-10—are being investigated for their ability to restore balance within the CNS [40,41]. This nuanced approach aims to modulate, rather than eliminate, neuroinflammation, reflecting the complexity and importance of maintaining immune equilibrium in the brain.

Recent studies have emphasized the dual role of microglia and astrocytes as both protectors and mediators of damage in conditions such as Alzheimer’s disease, Parkinson’s disease, and aging-related neurodegeneration. Table 1 provides a concise overview of key findings from the literature, highlighting how these cellular players and molecular pathways shape the immune landscape within the CNS and their implications for therapeutic targeting. The studies included in Table 1 were selected based on the specific following criteria: (1) relevance to core neuroimmune processes, (2) the inclusion of high-impact and peer-reviewed publications from the last 5 years (with exceptions for foundational studies), (3) the utilization of rigorous methodologies such as transcriptomic analysis, in vivo imaging, and cytokine profiling, and (4) direct applicability to therapeutic advancements for CNS diseases. The limitations of each study are also noted to ensure a balanced perspective.

### 2.4. Emerging Theories and High-Impact Concepts in Neuroimmune Interactions

As neuroimmunology progresses, several newly established theories are reshaping our understanding of the CNS’s immune landscape and its impact on neurological health. These concepts not only provide fresh insights, but are also driving high-impact research within the field.

The Microglial “Memory” Hypothesis and Immune Priming: Emerging research suggests that microglia, the CNS’s primary immune cells, may retain a form of immune memory through epigenetic modifications, a hypothesis that has significant implications for neurodegenerative diseases. Unlike peripheral immune cells, which retain memory to respond rapidly to previously encountered pathogens, microglial memory could have a more mixed impact within the CNS [47]. Studies reveal that, after an initial inflammatory event, microglia can become “primed”, maintaining a heightened state of responsiveness that allows them to overreact to subsequent minor triggers [48]. This primed state is thought to involve epigenetic alterations, such as changes in DNA methylation and histone modification, which persist within microglia long after the initial stimulus has been resolved. In neurodegenerative diseases like Alzheimer’s and Parkinson’s, this priming may drive chronic neuroinflammation, as primed microglia respond excessively to Aβ or α-synuclein accumulation, exacerbating neuronal damage [49]. Therapeutic strategies that specifically target primed microglia or reset their epigenetic landscape are now being explored as potential interventions to prevent or slow the progression of neurodegeneration [47].

The Gut–Brain Axis and Neuroimmune Crosstalk: Increasingly, evidence supports the concept of a bidirectional communication pathway between the gut microbiome and the CNS, known as the gut–brain axis [50]. Recent studies highlight that gut-derived immune signals can influence microglial activity and CNS inflammation, establishing a link between peripheral immune health and neuroimmune responses. The presence of certain microbial metabolites in the bloodstream, such as short-chain fatty acids (SCFAs), has been shown to modulate microglial activation and affect neuroinflammatory responses [51]. Disruptions in the gut microbiome, often characterized by a reduced microbial diversity and changes in SCFA production, are associated with an increased risk of neuroinflammatory diseases [52]. This connection suggests that the gut microbiota could play a significant role in modulating neuroimmune activity, and therapies aimed at restoring gut health may hold potential as complementary treatments for CNS disorders. The gut–brain axis concept has opened up new avenues for investigating how dietary interventions, probiotics, and even fecal microbiota transplants could influence CNS immune health [53].

Peripheral Immune Cell “Education” by the CNS and Neuroimmune Conditioning: Another emerging theory posits that peripheral immune cells may undergo a form of “education” through interactions with CNS-resident immune cells and molecules [54]. This process, referred to as neuroimmune conditioning, suggests that immune cells exposed to the CNS environment may adopt specialized functions when they re-enter the peripheral circulation. Studies have shown that certain T cells, after passing through the CNS, acquire properties that allow them to interact with microglia more effectively upon subsequent entry [55]. This conditioning process may help to “prime” the peripheral immune system for future CNS insults, potentially aiding in faster responses to infection or injury. However, this conditioning also introduces risks; in autoimmune conditions like multiple sclerosis, these CNS-conditioned T cells may contribute to excessive immune activity within the CNS, perpetuating inflammation [56]. Understanding the mechanisms of neuroimmune conditioning could offer new insights for designing therapies that modulate T cell responses to CNS antigens, aiming to reduce harmful autoimmunity while preserving beneficial immune surveillance [57,58].

The “Dark Side” of Immune Checkpoint Molecules in the CNS: Immune checkpoint molecules, traditionally studied in the context of cancer immunotherapy, have recently garnered attention for their roles in the CNS, particularly in regulating microglial activity and neuronal survival. Molecules such as PD-1 and CTLA-4, which act as brakes on immune responses in peripheral immunity, are also expressed by CNS-resident immune cells and have been implicated in neuroimmune regulation [59,60]. In cancer, blocking immune checkpoints like PD-1 has led to breakthrough therapies that enhance T cell activity against tumors. However, in the CNS, the role of immune checkpoints is more complex [61]. Checkpoint molecules in the brain appear to maintain a protective role, helping to prevent excessive microglial activation and neuronal damage. Studies have shown that the loss of PD-1 or CTLA-4 activity in the CNS can exacerbate neuroinflammation and neurodegeneration, as unchecked microglial activation leads to greater oxidative stress and cytokine production [62]. Thus, the “dark side” of immune checkpoints in the brain highlights a cautionary tale for therapies that might inadvertently disrupt these regulatory systems [63]. New research is exploring ways to harness immune checkpoints to achieve neuroprotection, potentially using modified checkpoint therapies that act specifically within the CNS without compromising systemic immune function [64,65].

Interference with the BBB as a Controlled Therapeutic Strategy: Recent developments in BBB research are redefining our approach to CNS drug delivery. The BBB’s role as a restrictive barrier has historically limited treatment options for CNS diseases, but controlled modulation of the BBB now appears feasible [66,67,68]. Technologies such as FUS, in combination with microbubbles, have shown promise in transiently and selectively opening the BBB, allowing for the targeted delivery of high-molecular-weight drugs, antibodies, or gene therapies [69]. This controlled BBB modulation could enable the administration of therapeutic agents that were previously unable to penetrate CNS tissue, revolutionizing treatment approaches for neurodegenerative diseases and brain cancers. Moreover, advancements in nanotechnology are further enhancing this strategy, with nanoparticles being engineered to carry drugs across the BBB via receptor-mediated transcytosis [70,71]. By temporarily and selectively increasing BBB permeability, clinicians may achieve localized drug delivery to diseased areas without broadly disrupting CNS integrity. These emerging BBB modulation methods represent a new era in neuroimmune-targeted therapies, offering both precision and access in CNS drug delivery [72].

Each of these high-impact concepts is reshaping neuroimmunology, highlighting the field’s evolving landscape and the intricate, bidirectional influence between immunity and neural function. By delving into these cutting-edge areas, this work offers a comprehensive view of the complexities and potential within neuroimmune interactions, underscoring the promise of innovative, precisely targeted therapies for CNS disorders.

## 3. Neuroimmune Mechanisms in CNS Tumors and Cancer Progression

### 3.1. Neuroimmunology of CNS Tumors

CNS tumors, particularly high-grade gliomas like glioblastomas, present a fascinating paradox in immunology. While these tumors do recruit immune cells, they transform the immune landscape into a fortress that supports, rather than opposes, tumor growth [73]. This transformation, often called immune “re-education”, renders immune cells like tumor-associated macrophages (TAMs) and resident microglia complicit in the tumor’s defense, turning them into allies that secrete immunosuppressive cytokines such as IL-10 and TGF-β [74,75]. These cytokines play a crucial role in muting the broader immune response, shielding the tumor from surveillance.

A breakthrough area of research involves exosomes—tiny extracellular vesicles released by glioma cells—as carriers of immunosuppressive signals. Recent studies suggest that exosomes from gliomas are loaded with microRNAs and other molecules that program immune cells even before they reach the tumor [76]. For instance, miR-21 and miR-146a within these exosomes can dampen immune cells, preparing them to support the tumor upon arrival [77]. This “pre-conditioning” strategy extends the tumor’s influence beyond its immediate environment, reshaping even the peripheral immune landscape [78].

Immune checkpoint molecules like PD-L1, which are known for blocking immune activity in other cancers, add another layer to gliomas’ defensive strategy [79]. PD-L1 is found not only on glioma cells themselves, but also within exosome cargo, forming an immune blockade that can impact both nearby and distant immune cells. By using PD-L1 in this way, gliomas effectively “turn off” T cells, creating a profoundly immunosuppressive environment [80]. Researchers are now exploring ways to target these immunosuppressive mechanisms—particularly the release of exosomes and the PD-1/PD-L1 pathway—to dismantle this multi-layered defense system [81].

Additionally, regulatory T cells (Tregs) and myeloid-derived suppressor cells (MDSCs) contribute to this tumor-protective environment. Gliomas actively recruit these immune cell types through chemokines like CCL22, which specifically attract Tregs [82]. These Tregs further silence the immune response, building an immune-tolerant “safe zone” that allows the tumor to flourish. New research is focused on disrupting these recruitment pathways or selectively blocking the suppressive functions of Tregs and MDSCs within gliomas, potentially opening up avenues for therapies that strip away the tumor’s protective shield and allow for a more effective immune attack [83]. Emerging research has elucidated mechanisms such as exosome-mediated suppression, checkpoint molecule expression, and epigenetic reprogramming, which are integral to glioma progression and resistance to therapy. Table 2 provides a detailed overview of these mechanisms, highlighting recent advancements in understanding glioma immunology and potential therapeutic targets to disrupt the tumor’s immunosuppressive strategies.

### 3.2. Immunotherapies in Neuro-Oncology

Immunotherapy has sparked excitement in cancer treatment, and CNS tumors are no exception. Yet, established approaches like checkpoint inhibitors and CAR-T cell therapies face unique challenges within the CNS. Checkpoint inhibitors, which target proteins like PD-1/PD-L1 and CTLA-4 to release “brakes” on immune cells, have shown remarkable success in many cancers, but often encounter resistance within gliomas [89,90]. One barrier is immune exclusion—the limited presence of cytotoxic T cells in the glioma microenvironment, which reduces the efficacy of these inhibitors. To address this, researchers are developing combination therapies that pair checkpoint inhibitors with agents designed to attract T cells to the tumor or to break down local immunosuppressive barriers. For example, recent trials combining anti-PD-1 inhibitors with oncolytic viruses have shown potential [91,92]. These viruses selectively infect tumor cells and create localized inflammation, which attracts T cells and establishes an environment that amplifies the effects of checkpoint inhibition.

CAR-T cell therapy, a promising alternative, is being customized for glioblastomas by targeting specific mutations like EGFRvIII, found only on glioblastoma cells. These CAR-T cells can selectively seek out and destroy EGFRvIII-expressing cells, sparing normal tissues [93]. Yet, challenges remain—limited CAR-T cell access across the BBB and the presence of glioma cells that lack the target antigen. Researchers are exploring dual-target CAR-T cells that can recognize multiple glioma-associated antigens, which may reduce the risk of tumor cells escaping detection and provide a more comprehensive therapeutic response [94].

In an exciting advancement, CAR-T cell designs now feature molecular “switches” that allow for precise control over CAR-T cell activity within the CNS. These switches can be activated or deactivated in response to specific signals, offering the flexibility to modulate CAR-T cell responses and reduce neurotoxic side effects [95]. Such control mechanisms are particularly valuable in the brain, where uncontrolled immune responses can cause significant inflammation and damage. Early studies suggest that these “switchable” CAR-T cells may provide a safer, more adaptable form of therapy for brain tumors, allowing for targeted treatment with minimized risks [96].

### 3.3. Neurological Complications of Systemic Cancer Immunotherapy

The rise of systemic immunotherapies, especially immune checkpoint inhibitors, has illuminated a new set of neurological complications—immune-related adverse events (irAEs), which can affect the CNS. These complications range from mild symptoms, like headaches, to severe conditions such as encephalitis and Guillain–Barré-like syndromes [97]. Studies suggest that some irAEs result from immune cross-reactivity, where immune cells activated by cancer therapy mistakenly attack neural tissues due to antigen similarities [98]. This phenomenon highlights the fine balance between robust cancer-targeting immune activation and the risk of unintended attacks on CNS tissues [99].

Efforts to predict and prevent these irAEs are advancing, with a particular focus on identifying biomarkers that may indicate a patient’s susceptibility to CNS irAEs [100]. Elevated levels of cytokines like IL-6 and TNF-α before treatment, for example, have been linked to a higher risk of CNS complications, pointing to the potential for pre-treatment cytokine profiling [101]. In patients who do experience irAEs, low-dose corticosteroids are commonly administered to reduce inflammation, though this can also diminish the primary immunotherapy’s effectiveness. To address this challenge, researchers are developing biologics that target specific inflammatory pathways, aiming to reduce neurotoxicity without broadly suppressing immune function [102].

In CAR-T therapies, new safety mechanisms are being integrated to manage neurotoxicity. Pharmacological agents are now being used to temporarily suppress CAR-T cell activity in cases of severe neurotoxic symptoms [103]. These “safety valve” systems allow for rapid adjustment, ensuring the protection of neural tissues during intense immune responses while maintaining the treatment’s overall effectiveness [104]. This careful modulation represents a significant advancement in the safety of CAR-T cell therapy, allowing a balance between powerful cancer treatment and protection for the CNS [105].

## 4. Inflammation and Cerebrovascular Diseases

### 4.1. Role of Inflammation in Stroke Pathogenesis

Inflammation is now seen as a profound contributor to the risk, progression, and aftermath of stroke, entwining systemic and localized processes that prepare the ground for ischemic events. Chronic inflammation, marked by high levels of IL-6, TNF-α, and C-reactive protein (CRP), creates a landscape ripe for thrombosis, transforming vascular health and priming atherosclerotic plaques for rupture [106]. In this setting, the actions of neutrophils are particularly compelling; recent findings have shown that they form dense webs of neutrophil extracellular traps (NETs), which capture platelets and other clotting factors, setting up a microenvironment ideal for clot formation [107].

On a cellular level, the neurovascular unit—a delicate consortium of endothelial cells, astrocytes, neurons, and pericytes—becomes a theater of inflammatory action. Here, inflammatory molecules compromise the unit’s tightly regulated integrity, destabilizing the BBB and promoting infiltration by circulating immune cells. Molecular signals, like those from the NF-κB pathway, amplify this damage, prompting endothelial cells to express adhesion molecules that tether immune cells to the vessel walls [108]. As immune cells move into the CNS, they release enzymes that degrade the extracellular matrix, making the neurovascular architecture fragile and primed for injury [109]. This cellular choreography of inflammation and immune infiltration is fast becoming a target for therapies designed to strengthen the neurovascular unit’s resistance to inflammatory assault before a stroke even occurs [110].

Beyond these cellular mechanisms, systemic contributors such as acute and chronic bacterial or viral infections, as well as vaccination-related inflammatory responses, further amplify the risk of stroke by driving systemic inflammation. Chronic bacterial infections, such as *Helicobacter pylori* and periodontitis, trigger a persistent inflammatory state characterized by elevated levels of cytokines like IL-6 and TNF-α, along with acute-phase proteins such as CRP. This pro-inflammatory milieu exacerbates endothelial dysfunction, accelerates atherosclerotic plaque formation, and primes plaques for rupture, significantly increasing the likelihood of ischemic events.

Similarly, viral infections, including herpes simplex virus, cytomegalovirus, and influenza, have been linked to heightened stroke risk. These infections stimulate systemic immune activation and induce endothelial damage, either directly through viral replication or indirectly via immune-mediated responses [111]. For example, influenza infections are known to transiently elevate stroke risk due to increased systemic inflammation and hypercoagulability [112].

In contrast to infections, vaccinations may have dual effects. While vaccinations, such as those against influenza, are protective and prevent infection-driven inflammatory cascades, they can occasionally trigger mild, transient inflammation. In rare cases, this post-vaccination response may exacerbate underlying vascular conditions, especially in individuals with pre-existing risk factors [113].

These systemic triggers converge on shared molecular pathways, such as the activation of the NF-κB signaling cascade and an increased production of ROS. This results in endothelial activation, the upregulation of adhesion molecules, and the infiltration of immune cells into the neurovascular unit, amplifying local inflammation and structural fragility. Recognizing the roles of these systemic contributors highlights the importance of managing chronic infections and implementing preventative strategies, such as vaccinations, to mitigate inflammation-driven stroke risk [114].

### 4.2. Post-Stroke Immune Modulation

After a stroke, the body’s immune response proceeds in carefully orchestrated stages, each marked by unique cellular and molecular signatures. The initial hours see a surge of damage-associated molecular patterns (DAMPs) from dying neurons, which trigger microglia within the brain and monocytes from the bloodstream [115]. Microglia—activated like never before—release cytokines such as MCP-1 and IL-1β that beckon peripheral immune cells to the site of injury, where they work to clear debris and prevent infection. Yet, this acute inflammatory response, if left unchecked, can quickly expand the damage zone, pushing the brain into secondary injury territory [116].

Within days, however, the body initiates a dramatic shift from hyperinflammation to a phase of immunosuppression. Known as “stroke-induced immunodepression syndrome” (SIDS), this stage is a physiological pivot aimed at limiting ongoing inflammation within the brain [117]. The sympathetic nervous system releases norepinephrine, prompting a cascade of immune alterations that suppress T cell function and redirect lymphocytes back to lymphoid organs. This downshift in immunity is double-edged; it reduces the risk of further CNS injury, but renders patients highly susceptible to infections like pneumonia and urinary tract infections. Understanding the delicate mechanisms that dictate this immunosuppressive switch could reveal therapeutic pathways to more selectively modulate post-stroke immunity, preserving protection while reducing vulnerability [118].

The progression and severity of stroke-induced immunodepression syndrome are influenced by several interconnected factors. Overactivation of the sympathetic nervous system plays a critical role, as excessive norepinephrine release suppresses the activity of T cells and NK cells, significantly weakening the immune response to pathogens. This suppression, though initially protective against further inflammation in the brain, leaves the body defenseless against infections. Simultaneously, elevated levels of glucocorticoids such as cortisol further amplify these immunosuppressive effects. These stress hormones inhibit the production of pro-inflammatory cytokines and impair the recruitment of immune cells to infection sites, creating a high-risk environment for secondary infections [119].

Pre-existing health conditions also significantly contribute to the severity of post-stroke immunosuppression. Advanced age, diabetes, and chronic inflammatory diseases compromise baseline immune resilience, exacerbating the suppressive effects of SIDS. Moreover, the gut microbiota play a crucial role in systemic immunity. Stroke-induced disruptions to the gut microbiome, known as dysbiosis, are characterized by a reduced microbial diversity and a decreased production of SCFAs. This disturbance weakens the gut’s barrier function and allows harmful bacteria to translocate into the bloodstream, further heightening infection risk [120].

Peripheral immune cell exhaustion is another key factor that compounds the effects of immunosuppression. After a stroke, immune cells such as lymphocytes and monocytes often exhibit a reduced functional capacity and diminished cytokine production. This exhaustion is particularly pronounced in patients with pre-existing immune deficiencies, delaying recovery and increasing the likelihood of complications [121].

Understanding these factors provides a foundation for designing targeted therapeutic strategies. Interventions that modulate sympathetic nervous system activity, restore the gut microbiota balance, and rejuvenate exhausted immune cells hold promise for mitigating the harmful effects of stroke-induced immunosuppression. Such approaches could preserve the protective benefits of the immunosuppressive phase while reducing its detrimental consequences, ultimately improving recovery outcomes for stroke survivors [122].

### 4.3. Therapeutic Interventions Targeting Inflammation

The complexity of post-stroke inflammation has spurred the development of multi-pronged therapeutic strategies, each seeking to refine immune responses to optimize recovery. One focal point is the IL-1 pathway—a central coordinator of the acute inflammatory response. Clinical trials using IL-1 receptor antagonists have demonstrated promising outcomes, with a reduced infarct volume and improved neurological function in early studies [123]. Another approach targets matrix metalloproteinases (MMPs), enzymes that degrade the BBB and allow for harmful immune cell infiltration. By inhibiting MMPs, particularly MMP-9, researchers aim to preserve BBB integrity, limiting secondary damage and protecting brain tissue from inflammatory degradation.

Among the most intriguing areas of research is microglial polarization. Microglia exhibit the following two predominant states: a pro-inflammatory (M1) phenotype that releases cytotoxic molecules, and an anti-inflammatory (M2) phenotype that promotes healing. By shifting microglia from M1 to M2 profiles, researchers hope to leverage the brain’s innate capacity for repair [124]. Certain compounds, such as PPAR agonists, show promise in achieving this shift, offering a novel pathway to reduce brain injury while fostering a recovery-oriented environment [125].

Checkpoint inhibition, an innovation in cancer therapy, is now being examined for its potential in stroke. Immune checkpoint molecules, such as PD-1 and PD-L1, could offer a method to fine-tune immune responses, enhancing the activity of the T cells involved in repair without tipping the balance toward damaging inflammation. Early models have shown that modulating these pathways can sustain a protective immune presence in the post-stroke brain, promoting resilience in the recovery phase [126,127].

Stem-cell-based therapies offer yet another layer of innovation, providing not only cellular support, but also immune modulation. Mesenchymal stem cells (MSCs), in particular, secrete a range of anti-inflammatory cytokines and trophic factors that create a regenerative environment [128]. MSCs release exosomes loaded with regulatory RNAs and proteins, influencing immune activity in ways that favor recovery and limit secondary damage [129]. Early-phase trials indicate that MSCs can stabilize inflammation, reduce neural injury, and enhance recovery, making them a promising addition to the suite of post-stroke immunotherapies [130].

## 5. Neurodegenerative Diseases and Immunological Factors

### 5.1. Alzheimer’s Disease

Alzheimer’s disease (AD) research increasingly emphasizes neuroinflammation as a central element in disease progression. Microglia, the CNS’s resident immune cells, initially respond to Aβ plaques with phagocytic activity, but become maladaptive under prolonged exposure, entering a heightened inflammatory state. At the core of this response is the NLRP3 inflammasome, which, when activated by Aβ, triggers IL-1β secretion, driving inflammation that damages neurons [131,132]. Recent, a study using MCC950, a selective NLRP3 inhibitor, demonstrated significant reductions in IL-1β production and partial recovery in synaptic function, pointing to the potential of inflammasome-targeted treatments to counteract AD’s inflammatory component [133].

A new insight into AD’s pathophysiology involves the complement system, which regulates the microglial pruning of synapses. Normally, complement proteins like C1q aid in refining neural circuits by tagging weak synapses for removal [134]. However, in AD, these proteins are overexpressed, tagging functional synapses, especially in memory-associated regions [135]. Microglia then eliminate these synapses, a process correlated with memory loss. Complement inhibitors targeting C1q have shown potential in animal models by preserving synaptic networks and slowing cognitive decline, representing a promising intervention focused on maintaining neural connectivity [136].

Tau pathology, another critical aspect of AD, is increasingly being understood to drive inflammation through specific interactions with microglial receptors. The TREM2 receptor on microglia is essential for tau clearance, but patients with TREM2 mutations experience compromised microglial function, leading to tau accumulation [137,138]. In recent preclinical models, TREM2 agonists effectively enhanced microglial activity, facilitating tau removal and reducing neurofibrillary tangles by nearly 50% in targeted brain regions. These findings highlight TREM2 as a unique target for tau-specific interventions that could complement therapies directed at Aβ pathology [139].

### 5.2. Parkinson’s Disease

In Parkinson’s disease (PD), inflammation is intricately linked to the accumulation of α-synuclein aggregates (Figure 1), which disrupt neuronal integrity. These aggregates interact with microglia via Toll-like receptor 2 (TLR2), activating NF-κB, a key transcription factor that upregulates pro-inflammatory cytokines, amplifying neuronal damage. TLR2 inhibition has emerged as a viable approach; studies indicate that TLR2 antagonists reduce neuroinflammation and mitigate dopaminergic neuron loss in PD models, making it a focal area in therapeutic development [140].

The “gut–brain axis” hypothesis offers an additional layer to PD pathogenesis, suggesting that gut microbiota dysbiosis may initiate neuroinflammatory processes. Studies have linked an increased gut permeability and inflammation, driven by specific bacterial populations like Proteobacteria, to α-synuclein misfolding in the enteric nervous system [141,142]. This misfolded α-synuclein may then propagate through the vagus nerve to the brainstem. Clinical trials investigating fecal microbiota transplants (FMTs) and probiotic treatments aim to determine if modulating the gut microbiome could delay or even prevent PD progression by addressing this early inflammatory trigger [143].

In terms of neuroprotection, gene therapy delivering glial cell line-derived neurotrophic factor (GDNF) directly to affected brain regions shows potential. Traditional GDNF administration faced distribution challenges, but adeno-associated virus (AAV) vectors have now been shown to provide localized, long-term GDNF expression. A study reported that AAV–GDNF significantly preserved dopaminergic neurons, reduced α-synuclein pathology, and maintained motor function in PD models, underscoring its dual role in neuronal preservation and inflammation modulation [144].

### 5.3. Multiple Sclerosis

MS is being increasingly characterized by the dysregulation of B and T cell activity within the CNS, leading to recurrent demyelination and neuronal injury. B cells in MS play a critical role not only in antibody production, but also as antigen-presenting cells that sustain autoreactive T cell responses. Targeting CD20-positive B cells with monoclonal antibodies, such as ocrelizumab, has produced substantial improvements in MS patients by reducing their pro-inflammatory cytokine levels and relapse rates, with effectiveness in both the relapsing–remitting and primary progressive MS subtypes [145,146].

Recent research has also emphasized the role of Tregs in MS. These cells, which function to suppress autoimmune responses, are often numerically and functionally deficient in MS patients. Low-dose IL-2 therapy, designed to selectively expand Tregs, has shown early success, increasing functional Treg populations and decreasing relapse rates without causing broad immunosuppression. This Treg-based approach is emerging as a promising immune-modulatory strategy to balance protective and pathogenic immune activity in MS [147].

Newer strategies in MS treatment focus on active CNS repair, particularly remyelination. Drugs like clemastine promote oligodendrocyte precursor cell (OPC) differentiation, enabling remyelination around damaged axons [148,149]. Clinical trials with clemastine have demonstrated functional recovery in visual pathways, marking a shift in MS therapy toward restoring neural structures rather than solely limiting immune attacks [150]. The development of next-generation remyelination agents aims to build on these effects, representing a paradigm shift from inflammation control to direct neural repair [151].

Lastly, S1P receptor modulators offer the targeted control of immune cell migration. Drugs such as fingolimod trap lymphocytes in lymph nodes, preventing their entry into the CNS and reducing inflammation. The latest S1P modulators, including ozanimod, selectively target S1P receptor subtypes to minimize side effects, and trials indicate that they effectively reduce CNS lesions and slow disability progression [152]. This selective approach optimizes immune control while preserving systemic immune function, providing a balanced intervention for MS management [153].

In addition to these advances, established therapies like glucocorticosteroids and interferon beta remain foundational in the management of multiple sclerosis, particularly for relapsing–remitting forms of the disease. Glucocorticosteroids, such as high-dose intravenous methylprednisolone, are the treatment of choice for acute MS relapses. By suppressing immune cell migration across the blood–brain barrier and downregulating pro-inflammatory cytokines like TNF-α and IL-1β, they effectively shorten the duration and severity of relapse episodes. However, due to their systemic effects, the long-term use of glucocorticosteroids is limited by side effects such as osteoporosis and hyperglycemia, necessitating careful monitoring and alternative maintenance therapies [154,155,156].

Interferon beta, one of the earliest disease-modifying therapies (DMTs) for MS, continues to play a significant role in reducing relapse rates and slowing disease progression. This therapy exerts its effects by modulating immune responses, decreasing the activation and migration of autoreactive T cells, and promoting the production of anti-inflammatory cytokines such as IL-10 [157]. Its ability to reduce CNS lesions, as demonstrated through MRI studies, has made interferon beta a cornerstone treatment for relapsing–remitting MS. Despite its efficacy, tolerability issues, including flu-like symptoms and injection site reactions, have led to ongoing efforts to refine dosing regimens and delivery methods [158].

Together with these established therapies, modern approaches such as monoclonal antibodies, Treg expansion, remyelination agents, and S1P receptor modulators are reshaping MS management. While glucocorticosteroids and interferon beta address acute inflammation and relapse prevention, newer therapies aim to repair neural damage and optimize immune modulation, offering a multifaceted strategy to improve outcomes for MS patients.

A comprehensive understanding of neurodegenerative diseases requires a systems-level perspective, as these disorders share interconnected neuroimmune processes that dynamically shape disease progression. Microglial priming, astrocyte activation, and cytokine signaling represent central components of the neuroimmune response, with the maladaptive regulation of these elements contributing to chronic inflammation and neurodegeneration [159]. In Alzheimer’s disease, microglial activation via the NLRP3 inflammasome and complement system dysregulation highlight the dual role of microglia in both defense and damage, where failure to resolve inflammation accelerates synaptic loss and neuronal injury [160]. Similarly, in Parkinson’s disease, the accumulation of α-synuclein aggregates triggers microglia via TLR2 signaling, amplifying inflammatory responses, while dysfunction in gut–brain axis interactions suggests systemic contributions to CNS pathology [161].

Astrocytes play a pivotal role in bridging immune and neuronal responses across these diseases. In AD and PD, astrocyte-derived cytokines, including IL-6 and CCL2, promote immune cell recruitment and BBB disruption, creating a permissive environment for peripheral immune infiltration. Meanwhile, impaired astrocytic support of oligodendrocyte precursor cells (OPCs) in multiple sclerosis hampers remyelination, underscoring their role in both immune regulation and tissue repair [162].

Recent insights into the meningeal lymphatic system further integrate these processes, as the clearance of neuroinflammatory mediators, immune cells, and CNS-derived antigens relies on intact lymphatic drainage. Dysfunctions in meningeal lymphatics, alongside prolonged microglial and astrocyte activation, create a self-sustaining cycle of inflammation, promoting neurodegeneration across diseases such as AD, PD, and MS [163,164].

By viewing these neuroimmune components as interconnected rather than isolated processes, it becomes clear that future therapies must aim to restore immune balance across multiple levels. Targeting microglial priming, astrocyte dysfunction, and lymphatic clearance simultaneously—while incorporating precision delivery systems and patient-specific strategies—holds promise for disrupting the chronic inflammatory feedback loops that drive neurodegeneration. Advances in computational modeling and single-cell transcriptomics will further unravel these interactions, offering tools to visualize disease trajectories and guide interventions that promote long-term CNS resilience.

## 6. State-of-the-Art Neuromodulators and Therapeutic Approaches

### 6.1. Advanced Immunotherapies

Emerging immunotherapies for neurodegenerative diseases are underscoring new possibilities with precision-focused vaccines and tolerance-inducing techniques. In AD, novel vaccines like ACI-24 are designed to target Aβ deposits with an approach that avoids the complications of widespread immune activation [165]. By using liposomal nanoparticles to encapsulate Aβ, ACI-24 selectively prompts an immune response against the plaques without triggering T cells. In animal studies, this vaccine reduced Aβ plaque levels and provided early cognitive stabilization, demonstrating potential as a targeted treatment that limits unnecessary neuroinflammation [166,167].

MS is seeing advances through antigen-specific tolerance strategies that aim to reprogram immune cells to recognize myelin as “self”. Myelin peptides encapsulated within nanoparticles interact with dendritic cells to present these peptides to T cells in a non-inflammatory way, which promotes immune tolerance rather than attack [168]. This tailored approach significantly reduces the risk of relapses and lesion formation, making it a compelling alternative to traditional, broadly immunosuppressive therapies [169].

### 6.2. Neuromodulation Techniques

Refinements in neuromodulation have positioned techniques like deep brain stimulation (DBS), TMS, and vagus nerve stimulation (VNS) as promising therapies for complex neurological and neuroinflammatory disorders [170].

DBS, previously known for its effectiveness in movement disorders, is now being explored in AD. Stimulating the fornix—a region involved in memory—DBS has been shown to preserve the hippocampal volume and decrease the levels of inflammatory markers, including IL-6 and TNF-α. This dual effect of modulating both cognitive circuits and peripheral immune responses opens up new avenues for DBS as a therapy in AD, where inflammation and neurodegeneration are closely linked [171,172].

TMS, which uses magnetic fields to non-invasively stimulate specific brain areas, is being optimized for its effects on neuroplasticity and immune modulation. Recent studies targeting the prefrontal cortex with repetitive TMS (rTMS) have shown an increased release of brain-derived neurotrophic factor (BDNF), which supports synaptic function and neural health [173,174,175]. Additionally, TMS has been associated with decreased levels of systemic inflammatory cytokines in patients with treatment-resistant depression, highlighting its potential to target neuroinflammation in both mood disorders and neurodegenerative diseases [176].

VNS, which stimulates the vagus nerve to activate anti-inflammatory pathways, has been shown to reduce inflammation by controlling cytokine release through the cholinergic anti-inflammatory pathway [177]. Studies on autoimmune conditions have shown that VNS significantly decreases cytokine production from macrophages, and trials are now investigating its application in MS, where it may help to reduce systemic and CNS inflammation and lower relapse rates [178,179].

### 6.3. Novel Pharmacological Agents

Pharmaceutical advancements in neuroimmunology are focusing on therapies that effectively cross the BBB and act directly within the CNS, with bispecific antibodies, kinase inhibitors, and specialized small molecules leading these efforts.

Bispecific Antibodies: These antibodies are engineered to bind both transport receptors on BBB endothelial cells and disease-associated CNS proteins. In AD models, a bispecific antibody designed to target transferrin receptors on endothelial cells and amyloid plaques showed over a 50% reduction in amyloid load [180]. This approach enhances targeted delivery, effectively reducing amyloid burden and minimizing peripheral immune activation [181].

Kinase Inhibitors: Drugs targeting the kinases involved in neuroinflammatory pathways have gained traction for their ability to regulate microglial activity and inflammatory signaling. The JAK1/2 inhibitor baricitinib, originally approved for inflammatory arthritis, has shown promise in AD for its effects on IL-6 and IFN-γ reduction in the CNS [182,183]. Preclinical studies demonstrated improvements in cognitive function alongside reductions in neuroinflammatory markers, and ongoing clinical trials are assessing baricitinib’s impact on cognitive decline in AD patients [184,185].

BBB-Penetrating Small Molecules: Among small molecules, pioglitazone—a PPARγ agonist—has shown potential for its antioxidant effects and mitochondrial support within the CNS. In PD models, pioglitazone has been observed to reduce α-synuclein accumulation and protect neurons from oxidative stress [186]. Phase 2 trials have reported modest improvements in motor function and reduced CSF inflammatory markers, supporting the development of BBB-penetrating drugs that address neuroinflammation and mitochondrial dysfunction within the brain [187].

Table 3 highlights these cutting-edge strategies, summarizing their methodologies, key findings, and implications for advancing the treatment of CNS disorders.

### 6.4. Gene Therapy and Stem Cell Treatments

Gene and stem cell therapies represent two of the most advanced therapeutic avenues, offering the long-term modification of disease processes within the CNS.

AAV-Mediated Gene Therapy: Gene therapy using adeno-associated virus (AAV) vectors allows for the precise delivery of therapeutic genes directly into targeted brain regions. In PD, AAV vectors have been used to deliver GDNF, with results showing sustained neuroprotection and motor improvements. Recent trials have reported motor function stabilization in PD patients over a year following AAV–GDNF treatment, highlighting this gene therapy’s potential to provide durable benefits by enhancing the resilience of dopaminergic neurons [193,194].

CRISPR/Cas9 Genetic Editing: CRISPR/Cas9 technology is now being explored to edit genes associated with neurodegeneration. In AD, CRISPR has been applied to reduce the expression of the APP gene responsible for amyloid production. Early animal studies have shown marked reductions in amyloid burden and associated cognitive improvements, with treated models showing up to 60% fewer plaques and an enhanced memory performance [195,196]. This approach opens the door to disease-modifying interventions that target AD at its genetic roots.

Stem Cell Therapy Using MSCs and iPSCs: Stem cell therapies are advancing, particularly for diseases like MS and PD, where both immunomodulation and cell replacement are needed. Mesenchymal stem cells (MSCs) have shown success in reducing MS relapse rates by up to 50% through their anti-inflammatory effects and the secretion of growth factors that promote repair [197]. In parallel, induced pluripotent stem cells (iPSCs) are being developed to create patient-specific neuronal cells (Figure 2). In PD, iPSC-derived dopaminergic neurons have demonstrated the ability to integrate into brain tissue and restore dopamine function, while iPSC-derived oligodendrocytes in MS models have shown the potential for remyelination, providing hope for functional recovery in diseases characterized by demyelination [198,199].

## 7. Impact on Neurological Disease Management

### 7.1. Clinical Trial Advances

Recent clinical trials are breaking new ground in neurological disease treatment, bringing therapies that extend beyond symptom management to address the root causes of neurodegeneration, often with lasting effects [200,201].

PD: Gene therapies for PD are yielding remarkable insights, particularly in trials using AAV vectors to deliver GDNF directly to dopaminergic neurons. In a recent study, patients who received a single GDNF administration showed an increased dopamine transporter density in the substantia nigra and putamen, key areas affected in PD. These neuroprotective effects correlated with improved motor scores on the MDS-UPDRS scale, showing a sustained benefit over 18 months [202]. What is compelling is the observation that GDNF not only aids in neuron survival, but also appears to reduce inflammatory markers in the cerebrospinal fluid, including IL-6 and TNF-α [203]. This dual action suggests that GDNF could potentially alter disease progression on both the cellular and molecular levels, sparking new hope for PD treatments that address both neuroprotection and inflammation [204].

AD: The latest trials in AD are experimenting with combination therapies that simultaneously target Aβ and tau pathologies. One cutting-edge trial involves the use of bispecific antibodies, combined with tau kinase inhibitors, to slow down both amyloid deposition and tau propagation. Patients receiving this therapeutic “cocktail” experienced a 50% reduction in amyloid plaques, as shown by PET imaging, along with a marked decrease in tau levels in their cerebrospinal fluid [205]. Their cognitive scores on standardized scales, such as the Clinical Dementia Rating-Sum of Boxes, also showed stabilization, while advanced synaptic PET imaging with the tracer UCB-J indicated improvements in synaptic density. This multi-targeted approach not only promises to decelerate AD pathology, but also preserves synaptic health, providing a comprehensive intervention that addresses multiple facets of neurodegeneration.

MS: Antigen-specific tolerance therapies are a bold new direction in MS treatment, particularly for progressive forms of the disease [206]. A recent trial utilized a personalized approach, where dendritic cells (DCs) were engineered to present myelin antigens to the immune system in a way that induced tolerance rather than attack. Patients receiving these modified DCs showed a significant reduction in T cell responses to myelin antigens, while MRI scans indicated fewer gadolinium-enhancing lesions, a marker of acute inflammation. This trial also included MR spectroscopy, which revealed improved axonal health, as measured by N-acetylaspartate (NAA) levels, a marker of neuronal integrity [207]. This innovative approach could offer progressive MS patients a way to reduce immune attacks on myelin without the broader immune suppression that often brings unwanted risks.

### 7.2. Personalized Medicine and Biomarkers

The field of personalized medicine in neurology is rapidly evolving, thanks to advances in genetic profiling, wearable technology, and fluid biomarkers that allow for individualized therapies and the precise tracking of disease dynamics.

PD: Digital phenotyping, which combines genetic data with real-time monitoring via wearable devices, is revolutionizing PD management. In a recent study, patients with GBA mutations wore sensor-equipped devices that captured motor patterns and gait abnormalities in real time, enabling clinicians to optimize the doses of GCase modulators based on daily fluctuations in motor symptoms [208]. This personalized, dynamic dosing resulted in improved motor stability and a reduction in lysosomal dysfunction biomarkers, including cathepsin D, in the cerebrospinal fluid. The ability to adapt treatment based on daily biomarker feedback and genetic insights represents a shift towards highly responsive, individualized PD care that extends beyond static treatment regimens [209].

AD: Blood-based biomarkers, particularly the plasma levels of p-tau217 and Aβ42/Aβ40, are becoming essential for the early diagnosis and monitoring of AD progression [210]. A recent study found that elevated plasma p-tau217 accurately predicted amyloid pathology up to a decade before symptom onset, a finding that could enable earlier, pre-symptomatic interventions. Furthermore, advanced tau PET imaging using next-generation tracers like MK-6240 has demonstrated a high sensitivity in identifying tau accumulation in the earliest stages of AD [211]. These biomarkers provide a non-invasive window into disease progression, giving clinicians real-time tools to monitor and adjust treatments as patients respond to anti-amyloid and anti-tau therapies [212].

MS: Emerging biomarkers of neuroinflammation, such as YKL-40 and GFAP, are reshaping MS management, particularly for monitoring disease progression and treatment response. YKL-40, a marker of astrocyte activation, has been shown to correlate with MRI measures of lesion burden and white matter damage in MS, offering a real-time indicator of neuroinflammation [213]. Furthermore, studies reveal that YKL-40 levels can predict how patients respond to anti-CD20 therapies, allowing clinicians to select and monitor treatment more accurately based on baseline inflammation levels. This biomarker-guided approach minimizes exposure to ineffective treatments, enhancing both safety and efficacy in MS care [213].

Pharmacogenomics: Advances in pharmacogenomics are facilitating highly targeted treatments by considering genetic variations that influence drug metabolism and immune response. In MS, pharmacogenomic studies have shown that patients with polymorphisms in interleukin-7 receptor (IL-7R) genes respond more favorably to specific disease-modifying therapies, with fewer side effects [214]. This detailed understanding of genetic influence on drug efficacy enables clinicians to personalize therapy selection and dosing, increasing the likelihood of treatment success and reducing adverse events, especially in chronic and progressive conditions [215].

### 7.3. Challenges and Ethical Considerations

With the rise in advanced therapies in neurology come ethical, logistical, and financial challenges that require careful consideration to ensure equitable access and responsible application.

Cost and Access to Gene Therapies: The high cost of advanced therapies, particularly gene- and CRISPR-based treatments, creates significant challenges for accessibility. While these therapies offer potential one-time solutions, their upfront costs, often exceeding USD 400,000, make them unaffordable for many. A new proposal, “annuity-based pricing”, aims to spread these costs over several years based on outcomes, which may help to increase their accessibility. However, this model is complex to implement in diseases with slow progression, such as PD, where benefits may take years to fully manifest. Real-world studies are essential to evaluate the long-term impact of these payment models and establish metrics that reflect patient benefits, supporting the sustainable adoption of high-cost therapies.

Informed Consent for Gene-Editing Trials: Informed consent is especially crucial in trials involving irreversible genetic modifications, such as CRISPR-based gene editing. Given the potential for unintended, lifelong effects, recent trials have adopted a multi-phase consent model where patients are offered genetic counseling at every stage. This approach, currently used in AD trials involving CRISPR, ensures that participants receive comprehensive information on potential risks, including unforeseen genetic changes. Assigning each patient a dedicated genetic counselor helps to maintain their autonomy, as patients can make informed decisions with a thorough understanding of the potential long-term consequences of gene editing.

Data Privacy and Genetic Ownership: The reliance on genetic and biomarker data in personalized medicine brings significant concerns around data privacy and ownership. To address these issues, some trial frameworks are now implementing patient-owned data models, where individuals retain full control over their genetic information and decide on its usage and sharing. Additionally, the concept of “data escrow” is gaining traction, allowing patients to securely store their genetic data with a third-party service, granting access only to authorized healthcare providers. This model ensures patient control, prevents unauthorized data access, and safeguards privacy, which is essential as genetic data become a routine part of neurological care.

Long-Term Cost-Effectiveness and Adaptive Reimbursement: Determining the cost-effectiveness of high-cost therapies is critical, especially as healthcare systems face budgetary constraints. Adaptive reimbursement models are now being proposed, where payment is structured around real-world outcomes. For example, in stem cell therapy for progressive MS, costs could be adjusted based on observed improvements in functional metrics such as mobility or MRI-based markers of neuroprotection. This model aligns costs with tangible benefits, helping to ensure that resources are used efficiently while also encouraging innovation in the development of therapies with long-term impact.

## 8. Future Directions

### 8.1. Integrative Approaches Combining Immunology and Neuroscience

The fusion of immunology and neuroscience is opening up entirely new avenues for treating neurodegenerative diseases. Researchers are exploring ways to harness the immune system’s complex functions in the brain, seeking to design therapies that not only protect neurons, but also recalibrate immune responses that often exacerbate neurodegeneration [216,217].

Systems Biology and Computational Modeling: In the quest to understand diseases like Alzheimer’s and Parkinson’s on a deeper level, scientists are turning to systems biology and computational modeling. These advanced tools pull in massive amounts of data from genetics, protein behavior, and cellular interactions to build comprehensive models of disease mechanisms [218]. Using machine learning, these models can predict how subtle shifts in immune activity might either fuel or dampen neurodegeneration. In Alzheimer’s research, for instance, computational models have uncovered unexpected links between microglial activation and tau buildup, highlighting potential points of intervention [219]. This approach is also revolutionizing drug discovery, as models can simulate how new treatments might affect interconnected pathways, enabling faster, more targeted early-phase testing [220].

Precision Immunotherapy for Neurodegeneration: Personalized immunotherapy is now taking center stage, with therapies tailored to the specific immune markers of each patient. By analyzing unique neuroinflammatory signatures, researchers can identify patients who may benefit most from precise immune modulation. In MS, for example, studies have shown that patients with high levels of cytokines like IL-17 and IFN-γ respond better to targeted immunotherapies. Similarly, in Parkinson’s disease, trials are underway to target anti-inflammatory agents specifically in patients with elevated markers of microglial activation, a characteristic of faster disease progression [221,222]. Precision immunotherapy represents a highly individualized treatment model that aligns therapies with each patient’s unique immune landscape, which could dramatically improve treatment outcomes.

### 8.2. Innovations in Drug Development for Neurodegenerative Diseases

Drug development in neurodegenerative diseases is evolving rapidly, with researchers focusing on treatments that target the very roots of disease mechanisms. These innovative drugs extend beyond symptom relief, offering hope for true disease modification by tackling the underlying causes.

Novel Drug Classes Targeting Protein Aggregation: In diseases like Alzheimer’s and Parkinson’s, misfolded proteins such as amyloid-β, tau, and α-synuclein form harmful clumps that contribute to neurodegeneration [223]. Researchers have introduced a groundbreaking class of drugs known as “protein disaggregases”, designed to break up these protein clumps. One such drug, LMTX, is currently being tested in Alzheimer’s trials and has shown the ability to dismantle tau tangles in laboratory models [224]. Remarkably, studies have found that, as tau tangles diminish, inflammation markers drop as well, suggesting a ripple effect that reduces the immune response associated with neurodegeneration. This class of drugs holds promise for targeting not only Alzheimer’s, but also Parkinson’s and other conditions where protein misfolding plays a destructive role [225].

Anti-Inflammatory Agents with Blood–Brain Barrier Penetrance: New advances are helping anti-inflammatory drugs to reach the brain more effectively, which is essential in treating the neuroinflammatory component of diseases like Alzheimer’s. Baricitinib, a JAK1/2 inhibitor originally used in arthritis treatment, is now showing potential in Alzheimer’s trials by suppressing inflammatory cytokines directly within the central nervous system. In early studies, patients receiving baricitinib had lower CSF levels of IL-6 and TNF-α, two key players in neuroinflammation [226,227]. Another promising drug is nimodipine, a calcium channel blocker that has been shown to protect neurons from inflammatory damage by calming overactive microglia. These agents are part of a new approach to safely modulate brain inflammation without affecting the rest of the body, adding a layer of precision that was previously difficult to achieve [228].

Synaptic Protectors and Neurotrophic Agents: Protecting synapses—the connections that allow neurons to communicate—is a growing area of focus, as synaptic loss is often an early sign of neurodegeneration. BPN14770, a drug that enhances synaptic resilience, has shown potential in models of Alzheimer’s and Huntington’s diseases [229,230]. By increasing the levels of BDNF, it supports synaptic health and helps to maintain cognitive function. Researchers are also developing encapsulated versions of neurotrophic agents like BDNF to improve delivery to the brain, where these molecules can bolster neuronal health. These treatments provide a direct path to slowing synapse degradation, an approach that could delay cognitive symptoms before significant neuron loss occurs [231].

### 8.3. Vaccination and Prophylactic Immunotherapy for Neurodegeneration

Vaccination—a concept traditionally associated with infectious diseases—is now finding a place in the prevention of neurodegenerative disorders. These novel vaccines aim to neutralize the toxic proteins implicated in diseases like Alzheimer’s and Parkinson’s, preventing the accumulation of these harmful aggregates [232,233].

Vaccines Targeting Amyloid-β and Tau in Alzheimer’s Disease: Preventive vaccines targeting Aβ and tau are being developed to slow or even halt Alzheimer’s disease progression in individuals at a high risk. One candidate, ACI-24, is designed to stimulate the immune system to produce antibodies against amyloid plaques [234]. Using advanced liposomal nanoparticles, this vaccine delivers Aβ fragments that prompt an immune response to clear amyloid from the brain. Early studies show that ACI-24 reduces the amyloid burden in patients with mild cognitive impairment, hinting at its potential to stop Alzheimer’s in its tracks [235]. Another vaccine, focused on tau, aims to prevent the protein from tangling within neurons. Studies in animal models have shown that this tau-targeting vaccine preserves synaptic function and delays cognitive decline, representing a preventive option for those with genetic predispositions to Alzheimer’s [236,237].

Alpha-Synuclein Vaccines for Parkinson’s Disease: In Parkinson’s, α-synuclein proteins form toxic aggregates that damage neurons. Vaccines targeting α-synuclein, such as PD01A, are now undergoing trials to reduce these harmful deposits. PD01A uses synthetic α-synuclein fragments to train the immune system to recognize and clear abnormal proteins without affecting the healthy α-synuclein needed for normal cellular function [238]. This targeted immunization has shown promise in slowing motor symptoms in early-stage PD patients and could be an effective strategy for those with genetic mutations that heighten their risk. By preventing the buildup of toxic proteins, α-synuclein vaccines offer a proactive approach to protect high-risk individuals before significant neurodegeneration occurs [239].

mRNA Vaccination for Personalized Neuroprotection: Inspired by the success of mRNA technology in COVID-19 vaccines, researchers are developing mRNA vaccines for neurodegenerative conditions. Unlike traditional vaccines, mRNA vaccines can be tailored to each patient’s genetic profile, encoding specific fragments of tau, α-synuclein, or other disease-related proteins [240]. Early studies in animal models are exploring the use of mRNA to stimulate immunity against these proteins before symptoms arise. This personalized vaccine approach could be particularly valuable for individuals with familial histories of neurodegeneration, offering a preventive measure that could slow or halt the disease before it has a chance to progress [241].

### 8.4. Technological Innovations in Treatment Delivery

As the complexities of neurodegenerative diseases come to light, new technological approaches to treatment delivery are enhancing the precision, safety, and efficacy of interventions, especially in overcoming the challenge of targeting the brain directly.

Nanotechnology for Precise Drug Delivery Across the Blood–Brain Barrier: The BBB is a formidable obstacle in treating neurological diseases, but nanotechnology is offering ways to bypass this challenge. Specialized lipid nanoparticles, engineered to carry therapeutic molecules, can now cross the BBB and deliver drugs directly to affected brain regions. In recent experiments, nanoparticles encapsulating CRISPR-Cas9 have been used to edit genes linked to Alzheimer’s within neurons, reducing tau buildup in animal models [242,243]. By adding specific ligands to nanoparticle surfaces, researchers can increase targeting precision, allowing for localized drug release with a minimal impact on the rest of the body. This technology is reshaping the possibilities for delivering gene therapies and targeted drugs to treat Alzheimer’s, Parkinson’s, and beyond [244].

Biodegradable Implants for Long-Term Drug Release: Biodegradable implants are emerging as a promising option for sustained, localized drug delivery within the brain. These implants slowly dissolve, releasing therapeutic agents such as neurotrophic factors or anti-inflammatory drugs directly where they are needed. In a trial for Parkinson’s disease, biodegradable implants loaded with GDNF were implanted near dopaminergic neurons, providing sustained neuroprotection and motor function improvements over months [245]. This method reduces the need for frequent injections and limits systemic side effects, offering a controlled-release option that could enhance the quality of life for patients with neurodegenerative conditions [246].

Wireless Neuromodulation and Adaptive Deep Brain Stimulation: DBS technology is advancing with wireless and adaptive systems, transforming how we manage complex symptoms in diseases like Parkinson’s [247]. New DBS devices can now be programmed remotely, allowing clinicians to adjust their settings in response to changes in patients’ symptoms without requiring in-person visits. These adaptive systems use real-time brain feedback to adjust stimulation levels automatically, creating a more responsive and personalized treatment experience [248]. Early trials have shown that adaptive DBS provides better motor control and fewer side effects than traditional systems, improving the quality of life and accessibility for patients who live far from specialized care centers [249].

Focused Ultrasound for Non-Invasive Drug Delivery and Neuromodulation: FUS is a novel, non-invasive technology that opens up temporary, localized “windows” in the BBB, allowing for the direct delivery of drugs, antibodies, and even gene therapies into targeted brain regions [250]. This technique has been successful in glioblastoma trials, where it increased the efficacy of chemotherapy. Researchers are now exploring its potential for delivering neuroprotective agents in Alzheimer’s and Parkinson’s patients. FUS is also showing promise as a standalone treatment for reducing motor symptoms in PD by modulating neural circuits without the need for implants, offering a minimally invasive alternative for precise, effective neuromodulation [251].

## 9. Conclusions

### 9.1. Summary of Key Findings

The convergence of immunology and neuroscience is fundamentally reshaping our approach to neurodegenerative diseases, highlighting new pathways to alter disease progression, not just manage symptoms. A foundational shift is underway, focusing on therapies that recalibrate the immune system to support neural health, rather than simply suppressing immune activity. This emerging paradigm, which we term neuro-immune modulation, envisions the immune system as a potential ally, one capable of preventing and even reversing neurodegeneration. By fostering a balanced immune response within the CNS, future therapies could enhance neuronal resilience while addressing the complexities of immune regulation in the brain.

Across diseases like Alzheimer’s, Parkinson’s, and multiple sclerosis, we are seeing a promising convergence of precision therapies that target multiple disease pathways. The dual-pathology approach in Alzheimer’s disease, for instance, suggests that tackling Aβ plaques and tau tangles concurrently could more effectively slow cognitive decline than focusing on one pathology alone. Our work suggests that a multi-target strategy could serve as a foundational model for future Alzheimer’s treatments, encouraging simultaneous interventions to address the interconnected nature of protein aggregation, inflammation, and neuronal dysfunction.

Similarly, in Parkinson’s disease, the concept of sustained neuroprotection through gene therapy aligns with what we propose as the neurotrophic hypothesis of PD progression. This approach supports the idea that dopaminergic neuron survival depends on consistent neurotrophic support, which diminishes as the disease advances. Through gene therapies that restore this essential support, we can open up a pathway toward fundamentally changing the natural course of Parkinson’s, moving beyond symptom management to neuroprotection.

The advent of personalized medicine is also bringing powerful new dimensions to neurodegenerative care. By tailoring therapies to each patient’s unique genetic, molecular, and immunological profiles, we can apply biomarker-guided precision medicine, which redefines treatment standards. Biomarkers such as phosphorylated tau (p-tau) in Alzheimer’s and neurofilament light chain (NfL) in MS allow for real-time monitoring that can guide therapeutic adjustments, track treatment efficacy, and offer an ongoing view of disease progression. These precision approaches have the potential to transform neurological care, enabling dynamic, responsive interventions that address the specific disease signatures of each individual.

Preventive strategies, including vaccines targeting neurodegenerative proteins, offer a promising shift toward early intervention. Vaccines directed at tau in Alzheimer’s or α-synuclein in Parkinson’s aim to disrupt the early stages of protein aggregation long before symptoms manifest. This approach aligns with a pathogenic protein spread model, wherein the spread of misfolded proteins through neuronal networks amplifies neurodegeneration. Vaccination against these toxic proteins may offer a viable path to stopping disease progression in individuals at a high risk, signaling a new era of proactive, preventive neurodegenerative care.

### 9.2. The Road Ahead in Neuroimmunology Therapeutics

As we look forward, the path of neuroimmunology is guided by interdisciplinary research, precision treatment, and a commitment to refining therapies tailored to individual needs. The synergy between immunology and neuroscience offers us new avenues to tackle neurodegenerative diseases at a systems level, with the potential to disrupt the feedback loops between immune cells and neurons that have long fueled disease progression. Future work in systems neuroimmunology, harnessing computational models to map and predict these complex interactions, could transform our ability to visualize disease progression in real time, guiding more accurate and adaptive therapies.

Innovation in drug delivery will be central to future advancements, particularly as we harness technologies that allow therapies to reach the brain with a greater precision and fewer side effects. Nanotechnology, biodegradable implants, and FUS are making localized treatments possible, helping us to overcome the challenge of the blood–brain barrier. These delivery technologies enable targeted approaches to treat the brain effectively, setting the stage for new therapies in gene editing, protein modulation, and immune support.

Personalized and precision medicine will continue to sharpen our therapeutic approaches, refining the specificity with which we treat neurological disease. Pharmacogenomics will enable us to customize dosages and select therapies based on individual genetic profiles, reducing the risk of side effects and enhancing therapeutic effectiveness. With real-time patient monitoring through wearable technology and biomarker tracking, treatments will increasingly adapt as the disease progresses, allowing for a responsive, patient-centered approach to neurodegenerative care.

As these innovations take shape, ethical considerations will be essential in ensuring responsible and equitable care. Transparent, patient-centered communication, especially for genetic interventions, will be vital in maintaining trust and securing informed consent. Gene editing, such as CRISPR-based treatments, calls for rigorous ethical oversight, given the irreversible nature of such interventions. To ensure that groundbreaking therapies are accessible, outcome-based pricing models that tie reimbursement to patient success may offer a pathway to making advanced treatments available to all who need them.

The future of neuroimmunology holds remarkable promise for transforming our approach to neurodegenerative diseases. With an emphasis on prevention, precision, and adaptability, we can envision a future where neurological conditions are not simply managed, but fundamentally altered to preserve quality of life and cognitive function. The advancements unfolding today inspire optimism for tomorrow: personalized therapies that make a meaningful impact, an era where debilitating progression can be delayed or even prevented, and a world where individuals with neurodegenerative diseases can live healthier, more fulfilling lives.

While these innovations hold transformative potential, their clinical translation is not without challenges. Immunotherapies, for example, present risks of off-target effects, where immune responses inadvertently damage healthy CNS tissues. Strategies to enhance precision, such as nanoparticle-based delivery systems or localized administration techniques like intrathecal injections, are under development to limit systemic side effects and improve therapeutic specificity.

Another critical hurdle lies in the induction of immune tolerance, particularly for chronic treatments, where prolonged immune modulation can render therapies less effective over time. Combining immunotherapies with immune checkpoint modulators or cyclic dosing regimens can help to sustain their efficacy while minimizing immune exhaustion. Personalized immunoprofiling, which tailors treatments to individual immune responses, represents an emerging approach to overcoming these limitations.

Access to and the affordability of advanced therapies remain pressing concerns. Technologies such as BBB modulation, neuromodulation devices, and gene-based treatments often come with prohibitive costs, limiting their availability to broader patient populations. Efforts to scale up production, optimize delivery platforms, and introduce outcome-based reimbursement models tied to clinical success are key strategies to enhance affordability. Ensuring global accessibility will require expanding clinical trials to underrepresented regions and investing in portable, scalable technologies that can be integrated into diverse healthcare settings.

Furthermore, the safety and long-term monitoring of interventions, particularly those involving permanent genetic or immune modifications, are paramount. Therapies such as CRISPR/Cas9-based gene editing hold great promise but necessitate rigorous oversight to address potential off-target effects and unforeseen consequences. Robust post-treatment surveillance frameworks and transparent patient communication will be essential to building trust and ensuring responsible implementation.

By addressing these challenges, the emerging therapies in neuroimmunology can be refined to maximize their safety, efficacy, and accessibility. With continued innovation and a focus on overcoming these barriers, we move closer to translating cutting-edge research into practical solutions that transform the lives of patients with CNS disorders.

## Figures and Tables

**Figure 1 ijms-25-13614-f001:**
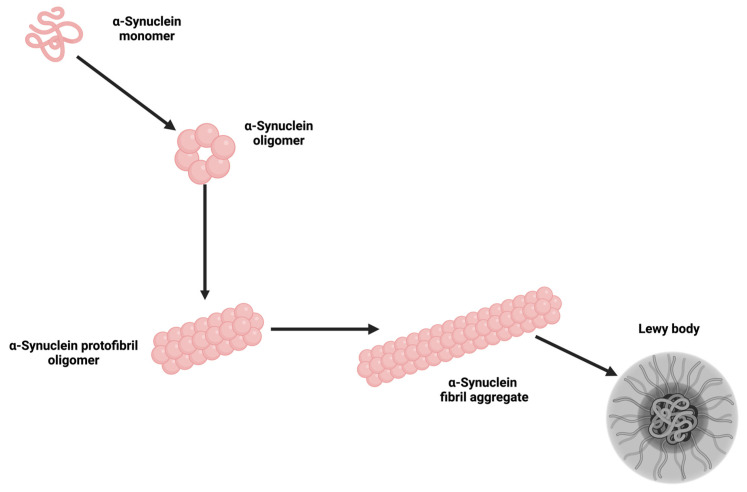
This figure depicts the aggregation pathway of α-synuclein in Parkinson’s disease.

**Figure 2 ijms-25-13614-f002:**
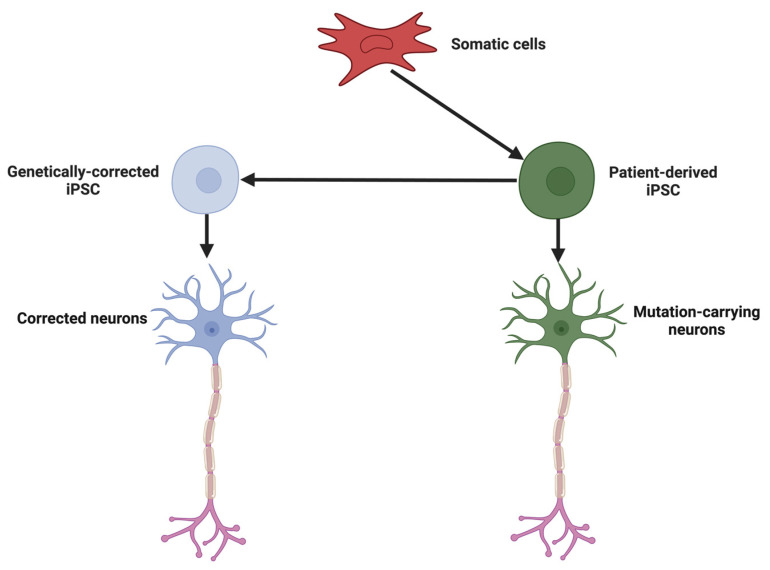
This figure illustrates the generation of corrected neurons from patient-derived induced pluripotent stem cells (iPSCs). Somatic cells are reprogrammed into patient-derived iPSCs, which can then be differentiated into mutation-carrying neurons. Genetic correction of the iPSCs results in genetically corrected iPSCs, which are subsequently differentiated into functional corrected neurons. This approach highlights the therapeutic potential of iPSC technology for neurological diseases caused by genetic mutations.

**Table 1 ijms-25-13614-t001:** Neuroinflammation and cellular interactions in the CNS. Abbreviations: CNS, Central Nervous System; BBB, Blood–Brain Barrier; IL, Interleukin; TNF-α, Tumor Necrosis Factor-Alpha; MMP, Matrix Metalloproteinase; TLR, Toll-Like Receptor.

Author(s)	Focus	Methods	Key Findings	Implications for Neuroimmunology	Limitations
Miao et al. [42]	Microglial response in Alzheimer’s disease	Transcriptomic analysis in AD mouse models	Discovered transcriptional changes that lead to microglial priming and chronic inflammation in response to amyloid plaques	Highlights microglia’s dual role in defense and neurodegeneration, suggesting microglial modulation as a therapeutic target	Limited to animal models; requires validation in human tissues
Schiera et al. [43]	Astrocytic cytokine release and BBB regulation	In vitro astrocyte culture; cytokine assays	Found that astrocytic IL-6 promotes BBB breakdown, exacerbating neuroinflammation	Emphasizes astrocytes’ role in CNS immune regulation, positioning them as potential therapeutic targets	In vitro model may not fully replicate in vivo complexity
Heidari et al. [44]	TLR4 signaling in Parkinson’s neuroinflammation	Mouse model of PD; TLR4 pathway inhibition	TLR4-mediated pathways increase inflammatory cytokines in response to α-synuclein, driving neuroinflammation	Positions TLR4 as a key target for modulating inflammation in PD	Does not address long-term effects of TLR4 inhibition
Luo et al. [45]	Astrocyte–microglia interactions during synaptic pruning	In vivo imaging in developmental models	Showed astrocyte cytokines direct microglial synapse pruning, affecting neural network formation	Links astrocyte-microglia signaling to neurodevelopment and potential early-life interventions	Requires longitudinal studies to assess developmental outcomes
Li et al. [46]	Chronic inflammation in aging microglia	Aging mouse models; cytokine profiling	Aging microglia exhibit high pro-inflammatory cytokines and reduced phagocytic function	Suggests aging as a factor in CNS immune dysfunction, relevant for Alzheimer’s and other neurodegenerations	Limited by lack of diverse age-related samples

**Table 2 ijms-25-13614-t002:** CNS tumor immunology and immunosuppression mechanisms. Abbreviations: CNS, Central Nervous System; PD-L1, Programmed Death-Ligand 1; Tregs, Regulatory T cells; MDSCs, Myeloid-Derived Suppressor Cells; TAMs, Tumor-Associated Macrophages; HDAC, Histone Deacetylase.

Author(s)	Tumor Type	Immune Modulation Mechanism	Methodology	Findings	Therapeutic Implications	Challenges
Zhou et al. [84]	Glioblastoma	Exosome-mediated immune suppression	Exosome analysis in glioma cell lines	Identified PD-L1 and microRNAs in glioma exosomes that reduce T cell activation	Targeting exosome release could dismantle glioma’s immunosuppressive network	Translational studies required to assess in vivo efficacy
Zhang et al. [85]	High-grade gliomas	PD-L1 expression in TAMs and microglia	Immunohistochemistry and PD-L1 assays	High PD-L1 levels in TAMs and microglia create immune tolerance zones around tumors	Supports PD-L1 as a checkpoint target to restore anti-tumor immunity	Limited efficacy in checkpoint blockade without T cell infiltration
Yang et al. [86]	Glioblastoma	Checkpoint inhibition and immune exclusion	Tumor microenvironment analysis; PD-1 inhibition trials	Immune exclusion limits checkpoint effectiveness in low-T cell gliomas	Encourages combination therapies to enhance T cell recruitment	Current combinations lack uniform effectiveness across patients
Wang et al. [87]	Gliomas	Epigenetic modulation of TAMs	HDAC inhibitor treatment on TAMs in glioma models	HDAC inhibition shifts TAMs toward a pro-inflammatory phenotype, enhancing tumor immunity	Suggests epigenetic reprogramming of TAMs as a viable immunotherapy approach for CNS tumors	Requires precise targeting to avoid off-tumor inflammation
Zhang et al. [88]	Glioma	Treg recruitment via CCL22 pathway	CCL22 blockade assays in mouse models	Blocking CCL22 reduces Treg infiltration, increasing cytotoxic T cell activity	Positions CCL22 as a target to reduce immunosuppression in gliomas	Need for more comprehensive studies on long-term Treg inhibition

**Table 3 ijms-25-13614-t003:** Novel immunotherapeutic and delivery strategies for CNS diseases. Abbreviations: CAR-T, Chimeric Antigen Receptor T-Cell Therapy; AAV, Adeno-Associated Virus; BBB, Blood–Brain Barrier; MSC, Mesenchymal Stem Cell; iPSC, Induced Pluripotent Stem Cell.

Author(s)	Therapeutic Strategy	Target Disease	Methodology	Key Findings	Clinical Implications	Limitations
Fletcher et al. [188]	Focused ultrasound (FUS) for enhanced BBB permeability	Glioblastoma	FUS with microbubbles in murine models	Safe, transient BBB opening enhanced drug delivery to glioma sites	Positions FUS as a viable BBB-targeting tool for precise CNS drug delivery	Requires refinement to avoid off-target BBB disruption
Li et al. [189]	Bispecific antibodies targeting PD-1 and CD47	Glioblastoma	Dual-targeting antibody trials in glioma models	Dual-target bispecific antibodies increase T cell activation and reduce immune evasion	Supports bispecific antibodies as an immunotherapy to counter glioma’s immune resistance	Risk of cross-reactivity with non-tumor cells
Dong et al. [190]	Switchable CAR-T cells targeting EGFRvIII	Glioblastoma	CAR-T design with molecular switches; EGFRvIII targeting	Switchable CAR-T cells reduce neurotoxicity and target EGFRvIII-positive cells	Proposes switchable CAR-T cells as safer brain tumor treatment	Potential antigen heterogeneity in gliomas reduces efficacy
Zeng et al. [191]	Oncolytic viruses for immune stimulation	Glioblastoma	Oncolytic virus administration and immune profiling	Increased T cell infiltration and sensitization to checkpoint inhibitors	Oncolytic viruses enhance immune responses, converting tumors into in situ vaccines	High potential for inflammatory side effects; needs regulation
Unnisa et al. [192]	Nanoparticle-assisted gene therapy for BBB penetration	Alzheimer’s Disease	Nanoparticle testing in AD models; BBB permeability assays	Nanoparticles successfully cross BBB and deliver therapeutic genes to neurons	Positions nanoparticle technology as a precise method for CNS-targeted gene delivery	Risk of nanoparticle accumulation in off-target sites

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
