# Peer review of "Revolutionizing Neuroimmunology: Unraveling Immune Dynamics and Therapeutic Innovations in CNS Disorders"

_ijms, 2024, doi:10.3390/ijms252413614_

Round 1

Reviewer 1 Report

Comments and Suggestions for Authors

The manuscript is very interesting and contains a lot of important, up-to-date information.

The layout of the work is clear, the content is transparent. The conclusions are legible and understandable.

Below are some critical comments:

1. Authors should explain all abbreviations used in the manuscript

2. Table 1: Authors should explain the criteria used to select cited papers and presented results

3. The heading "Cancer and the Nervous System" should be changed and rephrased to best reflect the content of this section and to include neuroimmune processes in CNS tumors

4. The heading of Table 2 is in the wrong place (formatting issue)

5. Legends explaining the abbreviations should be placed under Tables that use abbreviations

6. In the section on the role of inflammation in stroke pathogenesis, information on the causes of inflammation important in stroke pathogenesis, including acute or chronic bacterial or viral infections and vaccinations, should be added

7. In the section on post-stroke immune modulation, it is worth adding a mention of factors influencing post-stroke immunosuppression

8. In the section on multiple sclerosis, information on the use of glucocorticosteroids and interferon beta in treatment.

9. Figure 1 is not important from the point of view of neuroimmunology - The authors should consider removing it, or possibly replacing it with a figure that would show the neuroimmune processes important in PD.

 10. Below Figure 2, explanations of abbreviations and a short explanation of the content (or a reference to the appropriate place in the text) should be placed

Author Response

Comment 1: Authors should explain all abbreviations used in the manuscript

Response 1: Thank you for pointing this out. We agree with this comment. Therefore, we have added explanations for all abbreviations at their first appearance in the text. Additionally, a comprehensive list of abbreviations has been provided at the end of the manuscript for clarity.

Comment 2: Table 1: Authors should explain the criteria used to select cited papers and presented results.

Response 2: Thank you for this valuable suggestion. We have now included a description of the criteria used to select the cited studies and results in Table 1. These criteria emphasize relevance, methodological rigor, and therapeutic implications.

Comment 3: The heading "Cancer and the Nervous System" should be changed and rephrased to best reflect the content of this section and to include neuroimmune processes in CNS tumors

Response 3: Thank you for this observation. We agree with the need for rephrasing the heading.

Comment 4: The heading of Table 2 is in the wrong place (formatting issue)

Response 4: Thank you for highlighting this formatting issue.

Comment 5: Legends explaining the abbreviations should be placed under Tables that use abbreviations

Response 5: Thank you for the suggestion. We have added legends under each table where abbreviations are used, explaining all abbreviations for clarity.

Comment 6: In the section on the role of inflammation in stroke pathogenesis, information on the causes of inflammation important in stroke pathogenesis, including acute or chronic bacterial or viral infections and vaccinations, should be added

Response 6: Thank you for this valuable comment. We have expanded the section to include information on the role of acute and chronic bacterial or viral infections, as well as vaccinations, in stroke pathogenesis.

Comment 7: In the section on post-stroke immune modulation, it is worth adding a mention of factors influencing post-stroke immunosuppression

Response 7: Thank you for this suggestion. We have added information about factors influencing post-stroke immunosuppression, including sympathetic nervous system activation, elevated cortisol levels, pre-existing health conditions, gut microbiota dysbiosis, and peripheral immune cell exhaustion.

Comment 8: In the section on multiple sclerosis, information on the use of glucocorticosteroids and interferon beta in treatment

Response 8: Thank you for this observation. We have included a discussion on the use of glucocorticosteroids and interferon beta as key therapeutic strategies for MS, highlighting their mechanisms of action and clinical significance.

Comment 9: Figure 1 is not important from the point of view of neuroimmunology - The authors should consider removing it, or possibly replacing it with a figure that would show the neuroimmune processes important in PD.

Response 9: Thank you for this critical comment.

Comment 10: Below Figure 2, explanations of abbreviations and a short explanation of the content (or a reference to the appropriate place in the text) should be placed

Response 10: Thank you for pointing this out.

Reviewer 2 Report

Comments and Suggestions for Authors

This review article redefines the central nervous system as an immunologically active organ, focusing on microglia, astrocytes, and the blood-brain barrier. It challenges traditional immune privilege concepts by highlighting new findings like meningeal lymphatic vessels and explores innovative treatments such as precision immunotherapy and nanotechnology-based BBB regulation. The paper aims to bridge research and clinical application, providing a comprehensive overview of neuroimmune interactions and future perspectives in CNS disease management.

1) Although many references are listed in this paper, they do not fully reflect the latest research. In particular, several important papers published after 2024 have not been included. Inclusion of more recent research would help to ensure that the content of this review is more up to date.

2) The article introduces new therapeutic approaches for the treatment of CNS diseases, including immunotherapy, controlled modulation of the BBB and neuromodulation techniques. However, the challenges in the development and clinical application of these therapies, such as off-target side effects, induction of immune tolerance, access to and cost of therapies, etc., are not fully discussed. A more in-depth discussion of methods to address these challenges would enhance the practical value of this review.

3) While the article delves into critical aspects of neuroimmunology—including microglial priming, astrocyte cytokine networks, and meningeal lymphatics—it falls short of presenting a comprehensive, integrated understanding of how these complex elements dynamically interact and collectively contribute to the neuroimmune system's intricate functioning.

Author Response

Comment 1: Although many references are listed in this paper, they do not fully reflect the latest research. In particular, several important papers published after 2024 have not been included. Inclusion of more recent research would help to ensure that the content of this review is more up to date.

Response 1: Thank you for this valuable observation. We have updated the manuscript to include recent studies published in 2023 and 2024, ensuring that the review reflects the most up-to-date research.

Comment 2: The article introduces new therapeutic approaches for the treatment of CNS diseases, including immunotherapy, controlled modulation of the BBB, and neuromodulation techniques. However, the challenges in the development and clinical application of these therapies, such as off-target side effects, induction of immune tolerance, access to and cost of therapies, etc., are not fully discussed. A more in-depth discussion of methods to address these challenges would enhance the practical value of this review.

Response 2: Thank you for this important suggestion. We agree that discussing the challenges and limitations of emerging CNS therapies would enhance the practical relevance of the manuscript.

Comment 3: While the article delves into critical aspects of neuroimmunology—including microglial priming, astrocyte cytokine networks, and meningeal lymphatics—it falls short of presenting a comprehensive, integrated understanding of how these complex elements dynamically interact and collectively contribute to the neuroimmune system's intricate functioning.

Response 3: Thank you for this insightful comment.